# Triple-Layer Nanocomposite Membrane Prepared by Electrospinning Based on Modified PES with Carbon Nanotubes for Membrane Distillation Applications

**DOI:** 10.3390/membranes10010015

**Published:** 2020-01-16

**Authors:** Mohamed R. Elmarghany, Ahmed H. El-Shazly, Saeid Rajabzadeh, Mohamed S. Salem, Mahmoud A. Shouman, Mohamed Nabil Sabry, Hideto Matsuyama, Norhan Nady

**Affiliations:** 1Mechanical Power Engineering Department, Faculty of Engineering, Mansoura University, Mansoura 35516, EgyptMohamedsameh@mans.edu.eg (M.S.S.); m_shouman@mans.edu.eg (M.A.S.); Mnabil.sabry@gmail.com (M.N.S.); 2Chemical and Petrochemicals Engineering Department, Egypt-Japan University of Science and Technology (E-JUST), Alexandria 21934, Egypt; 3Research Center for Membrane and Film Technology, Department of Chemical Science and Engineering, Kobe University, Rokkodaicho 1-1, Nada, Kobe 657-8501, Japan; rajabzadehsaeid@gmail.com; 4Chemical Engineering Department, Faculty of Engineering, Alexandria University, Alexandria 21544, Egypt; 5Nanotechnology Center, Mansoura University, Mansoura 35516, Egypt; 6Polymeric Materials Research Department, City of Scientific Research and Technological Applications (SRTA-City), Borg El-Arab City, Alexandria 21934, Egypt

**Keywords:** membrane distillation, water desalination, polyethersulfone, three-layer nanocomposite membrane

## Abstract

In this work, a novel triple-layer nanocomposite membrane prepared with polyethersulfone (PES)/carbon nanotubes (CNTs) as the primary bulk material and poly (vinylidene fluoride-co-hexafluoro propylene) (PcH)/CNTs as the outer and inner surfaces of the membrane by using electrospinning method is introduced. Modified PES with CNTs was chosen as the bulk material of the triple-layer membrane to obtain a high porosity membrane. Both the upper and lower surfaces of the triple-layer membrane were coated with PcH/CNTs using electrospinning to get a triple-layer membrane with high total porosity and noticeable surface hydrophobicity. Combining both characteristics, next to an acceptable bulk hydrophobicity, resulted in a compelling membrane for membrane distillation (MD) applications. The prepared membrane was utilized in a direct contact MD system, and its performance was evaluated in different salt solution concentrations, feed velocities and feed solution temperatures. The results of the prepared membrane in this study were compared to those reported in previously published papers. Based on the evaluated membrane performance, the triple-layer nanocomposite membrane can be considered as a potential alternative with reasonable cost, relative to other MD membranes.

## 1. Introduction

Water is an unlimited renewable resource that is distributed on earth in liquid, solid, and vapor forms through a closed cycle (cyclo-hydrological) powered by the energy of the sun. Without water, no life on our blue planet is possible. Only 3% of all the water on the earth is available as potable water [1]. Furthermore, the need for freshwater has visibly increased during the past few years all over the world [2]. Desalination, which is a process of removing the salts present in water, represents the primary alternative source for freshwater. Typical thermal-based desalination techniques use intense energy and require considerable space and expensive equipment. Therefore, membrane-based desalination processes are preferred [3]. 

Membrane Distillation (MD) is a membrane-based separation technique that uses a hydrophobic porous membrane to separate warm saline feed and cold desalted condensate [4,5]. The membrane allows water vapor to pass through its pores due to the difference in vapor pressure across its sides. Hydrophobicity of the membrane retains liquid water from entering the membrane pores and flowing through its bulk structure. MD configurations include vacuum membrane distillation, sweeping gas membrane distillation, air gap membrane distillation, and direct contact membrane distillation [5].

MD exhibits many advantages over the conventional desalination techniques [6,7,8] such as lower energy consumption than that the multi-stage flash distillation, lower working pressure than that the Reverse Osmosis (RO) processes, low fouling phenomena compared to the RO desalination process [9,10,11] and almost 100% salt rejection. One of the major factors hindering MD development is the existence of membranes with a reasonable cost [12,13]. The commercial membranes that are suitable for MD are made of different polymers such as poly (vinylidene fluoride-co-hexafluoropropylene) (PcH), polytetrafluoroethylene (PTFE), polyvinylidene fluoride (PVDF), and polypropylene (PP) [14,15]. Although these polymers are hydrophobic by nature, which resulted in somehow high membrane performance by avoiding membrane wetting or capillary condensation during the water desalination, usually, they are expensive. Recently, more consideration has been given to investigate appropriate membrane materials suitable for the MD process to enhance the MD membrane durability for long-term applications by improving the membrane hydrophobicity [16,17].

One of the main techniques commonly used to enhance the membrane properties to be suitable for the MD process is blending with different nano-additives [18,19,20]. For example, the immense potential of carbon nanotubes (CNTs) as an additive has been a subject of interest to be used in the fabrication of the membranes [21,22]. Both single-walled carbon nanotubes (SWCNTs) and multi-walled carbon nanotubes (MWCNTs) have been included in different polymeric membranes to enhance their hydrophobicity [23,24]. SWCNTs have better properties than MWCNTs, but difficult dispersion is a noticeable issue. In addition to the known eminent hydrophobicity of the CNTs, the inclusion of CNTs increases the membrane surface roughness, which enhances the membrane surface hydrophobicity, a critical property for many MD applications [25,26,27,28].

Moreover, functionalized forms of CNTs have been used as an immobilized agent in the membrane for enhancing the membrane mechanical and thermal properties [29,30]. Also, modified graphene and graphene oxide (GO) as an additive used in the MD membrane preparation give a unique structure with superior mechanical stiffness, thermal stability, and considerable reduction in temperature polarization [31]. Hydrophobic nature, low cost, liquid water permeability, selective sorption of water vapors, and anti-fouling properties make modified graphene and graphene oxide membranes attractive for MD applications [32]. 

It is noteworthy that some polymers such as tetrafluoroethylene and hexafluoropropylene have recently attracted attention as a potential membrane material used for MD rather than PVDF due to the increased fluorine content resulting in higher hydrophobicity than the PVDF [31]. Similarly, poly (vinylidene fluoride-co-hexafluoropropylene) (PcH) [33] and poly (vinylidene fluoride-co-tetrafluoroethylene) (PVDF-TFE) [34] polymers are potential candidates for MD applications since the hydrophobicity of these membrane materials are remarkable (i.e., water contact angle > 154°).

Polyethersulfone (PES) is a vastly used polymer in the fabrication of microfiltration and ultrafiltration membranes [35,36]. It has many advantages compared to other polymers such as high thermo-plasticity, excellent mechanical properties, perfect chemical, and thermal stability at elevated temperatures, relatively higher transition temperature, and lower cost than other membrane materials that have been used for MD process [37,38,39]. Despite all the advantages of PES as membrane material, it is difficult to use this material for producing MD membranes because its hydrophobicity is relatively low. Composite PES membranes utilizing titanium oxide nanotubes as additives have been fabricated and used for vacuum membrane distillation and led to the formation of a highly dense layer of titanium dioxide on the membrane surface [40].

Electrospinning of polymer nanofibers has attracted considerable attention over the past decade as a direct and straightforward method of obtaining nanostructures for different applications [41]. The rapidly developing electrospinning technique is a unique way of getting new polymer nanofibers with a diameter, usually in the range of 50 nm to 500 nm. In the electrospinning method, a solution in which a material is dissolved in a solvent is used as a spinning material. There are some parameters related to the electrospinning, which can affect the morphology of the prepared nanofibers, and they are divided into three groups: solution characteristics, spinning environment, and spinning conditions [5,42,43]. Electrospinning with and without multi-nozzle systems have been used to fabricate nanofiber membranes for the applications of MD [33,44]. Electrospun nanofibers have been obtained using many hydrophobic polymers such as PVDF and PcH [45,46,47]. Compared to other known membrane fabrication methods such as phase separation, electro-spun membranes have unique characteristics such as small fiber diameter, high specific surface area, and high porosity [48].

To date, PES has rarely been applied for MD applications, as it is not inherently hydrophobic, although it has been widely used in porous membranes (MF and UF) preparation. Few studies investigated the utilization of phase-inversion-coated PES membranes in MD [49,50]. 

In this study, cost-effective material such as PES was used to prepare the bulk of the membrane by blending a tiny percentage (0.5%) of CNTs in its dope to obtain high porosity as well as acceptable hydrophobicity. Our initial evaluations showed that using only PES/CNTs as the membrane for MD applications is not suitable considering low liquid entry pressure (LEP), which means the membrane can be easily wetted by the feed solution. Electro-spun prepared PES/CNTs membrane was sandwiched between two layers of the PcH/CNTs as skin layers at both sides of the membrane bulk (top and bottom surfaces) to increase the MD performance by avoiding membrane wetting based on the increase in surface hydrophobicity and LEP. It was concluded that the product of the method used in this study combines the high total porosity of the membrane with high hydrophobicity and LEP of the membrane surface. The prepared MD membranes (Control PES/CNTs and PcH/CNTs and also triple-layer nanocomposite membrane) performance in terms of membrane water flux and salt rejection was determined for the desalting process by simulated saline water, and results were compared with those of the membranes prepared in similar studies.

## 2. Materials and Methods 

### 2.1. Materials

Polyethersulfone (PES) Ultrason E 6020P (glass transition temperature of Tg = 225 °C and a molecular weight of 58,000 g/mol) was obtained from BASF chemical company (Ludwigshafen, Germany). Poly (vinylidene fluoride-co-hexafluoropropylene) (PcH) in pellets form (glass transition temperature of Tg = 140 °C, density = 1.78 g/mL at 25 °C) was supplied by Sigma-Aldrich. 1-Methyl-2-Pyrrolidone (NMP) (>99% C5H9NO, and density of 1.028 g/mL) was purchased from CHEM-LAB, Belgium. N, N-Dimethylformamide (DMF) (HPLC grade, 99.8% and density of 1.4305 g/mL), N, N-dimethylacetamide (DMAc) (ACS, >99% and density of 0.937 g/mL) and acetone (HPLC grade, >99.9%) were purchased from DOP ORGANIK KIMYA (Turkey). Multi-walled carbon nanotubes (MWCNTs) (>90% carbon basis, a diameter of 110–170 nm, length of 5–9 µm and density of 1.7 g/L) were supplied by Sigma-Aldrich and used as it is.

### 2.2. Membrane Preparation

Triple-layer nanocomposite layer, as well as control membranes (pure PES and pure PcH), were prepared and their MD performance was investigated with the procedure mentioned as follows. A schematic diagram of the prepared membranes is shown in Figure 1. The CNTs content varied from 0.1 to 1 wt.%, and based on the optimal measured contact angles, 0.5% CNTs were chosen for the rest of the studies as this percentage results in high mechanical strength and hydrophobicity. The PES/CNTs solutions were prepared as follows: 0.5 wt. % multi-walled CNTs were dispersed in NMP: DMF solvent (volume ratio 1:9) by sonication (LABSONIC, model: LBS2 4.5 Lt, Falc Instruments, Treviglio, Italy) for 1 h. PES (23 wt.%) was added to the previously prepared suspension and kept under continuous stirring at 500 rpm for 12 h at 60 °C. Ultrasonic degassing for the dope solution was done for 1 h at room temperature (25 ± 2 °C). The pure PES membrane was prepared in the same manner but without adding multi-walled CNTs. Electrospinning was performed in the nanofiber electrospinning system (NANON-01A, Mecc Co., Ltd., Fukuoka, Japan). In the present study, the solution was first fed to the syringe (12.3 mm inner diameter), and a stainless-steel needle (OD: 0.9 mm, ID: 0.6 mm) was used. The distance between the needle tip and the collector was kept at 15 cm, an electric voltage of 20 kV was applied with a spinneret speed and width of 50 mm/s and 120 mm, respectively. The flow rate of the polymeric solutions was constant throughout the study at 1 cc/h, and the electrospinning time was 5 h. After that, the prepared membranes were put in an oven at 60 °C for 24 h to remove the residuals of the solvents. For the preparation of the PcH/CNTs membrane, 0.5 wt.% multi-walled CNTs were dispersed in the prepared mixture of solvents (DMAc: Acetone = 20:80 vol. %) by sonication for 1 h. PcH (15 wt.%) was added to the suspension and kept under continuous stirring at 500 rpm for 12 h. After that, ultrasonic degassing was done for 1 h for the solution at room temperature (25 ± 2 °C) before the electrospinning. Electrospinning was at 22 kV and feed rate of 1.2 cc/h with the same previous parameters. After electrospinning, the membrane was put in the oven at 60 °C for 1 day to remove the solvent residuals. The pure PcH membrane was prepared in the same manner but without adding the multi-walled CNTs. To prepare the triple-layer nanocomposite membrane, the previously prepared dope for each polymer was used as the same with 0.5 wt.% CNTs additive. During the electrospinning, a thin layer of PcH was prepared using 0.75 cc dope as an electrospun solution, followed by a thick layer of PES preparation using a 4-cc dope solution. Then, another thin layer of PcH was electrospun over the PES layer with the procedure mentioned above, and after that, the prepared membrane was kept at 60 °C for 24 h. The coded names of the prepared membranes with their composition of each prepared membrane are summarized in Table 1.

### 2.3. Membrane Characterization

#### 2.3.1. Membrane and Nanofiber Morphology

Transmission electron microscopy (TEM, 2100Plus, JEOL Ltd., Tokyo, Japan) was used to observe the distribution of the CNTs in the prepared membranes. The membrane cross-section sample was cut by using PowerTom Ultramicrotomes (RMS Boeckeler, Boeckeler Instruments Inc., Tucson, Arizona, USA) after hardness in Araldite^®^ Embedding Resin (Mollenhauer, Germany). Scanning electron microscopy (SEM) (JCM-6000PLUS NeoScope Benchtop SEM, JEOL Ltd., Tokyo, Japan) was used to examine the morphology of the surfaces of the fabricated membranes and to calculate the average diameter of the nanofibers of each membrane. SEM images for all prepared membranes were taken after coating the samples with Platinum/Palladium alloy.

#### 2.3.2. Liquid Entry Pressure (LEP)

Liquid entry pressure (LEP) of each membrane was examined using a simple test set, as shown in Figure 2. The membrane was placed in a special cell equipped with a pressure gauge, and a chamber was filled up with distilled water. The chamber was connected to an air cylinder by a throttling valve, which was used to gradually increase the air pressure over the free surface of distilled water. LEP was reported by starting with low pressure which was slightly increased until the liquid water penetrates the membrane.

#### 2.3.3. Membrane Morphology

The porosity (*ε*) of the fabricated membranes was obtained by measuring the wet and dry weights of the membrane samples. The wet weight of the membrane sample was measured after immersing it in ethanol for 10 min. After drying the sample, the dry weight of the sample was measured. The membrane porosity was determined using the following equation [46]:(1)$$\varepsilon =\;\frac{\frac{m_{w}-m_{d}}{\rho_{e}}}{\left[\frac{m_{w}-m_{d}}{\rho_{e}}\right]+\frac{m_{d}}{\rho_{P}}}$$
where $m_{w}$ is the wet membrane weight (g), $m_{d}$ is the dry membrane weight (g), $\rho_{e}$ is the density of ethanol (g/cm^3^) and $\rho_{P}$ is the density of the polymer or determined density of polymer/CNTs composite material (g/cm^3^).

ImageJ software was used to investigate the average pore size and the average fiber diameter of the prepared membranes [25,51,52]. In total, 400 pores were considered using SEM images to determine the average pore diameter of the membranes using image processing.

#### 2.3.4. Static Water Contact Angle

Contact angle measuring device (Drop Shape Analyzer—DSA100, KRÜSS GmbH, Hamburg, Germany) was used to determine the water contact angle for membranes samples. A water droplet of four microns was dropped over the surface of each membrane and was analyzed using the captured images. The reported value was the average of six readings on two different samples for each membrane.

#### 2.3.5. Membrane Thickness

The average membrane thickness was obtained through five measurements at different points on each membrane sample using a micrometer (range: 0–25 mm, precision: 2 µm, HDT, China).

### 2.4. Membrane Distillation (MD) Cell Design

The MD cell used in this study is a flat sheet design cell that was entirely fabricated at the workshops of Egypt-Japan University of Science and Technology (E-JUST). The cell consists of two identical parts made of aluminum sheets, as shown in Figure 3. The gross dimensions of the cell are 90 mm length by 90 mm width with a total height of 22 mm and inlet and outlet diameters of 6 mm for each part. The effective area of the membrane is 20.25 cm^2^, and the depth of the flow is about 2 mm. Rubber gaskets and membrane spacers (around 1.5 mm × 1.5 mm square pore size) are placed at each side of the membrane to avoid water leakage and to support the membrane, respectively.

### 2.5. Membrane Distillation Experiments

All direct contact MD experiments were conducted in a laboratory-scale experimental test rig consisting of a continuous hot loop at the feed side and a continuous lukewarm loop at the permeate side, as shown in Figure 4. The hot loop was heated via a water bath (type: SB-1000, Cole-Parmer, Tokyo RIKAKIKAI co., LTD) and circulated using a micro-pump (Micropump L20561 A-Mount Suction Shoe Pump Head; Cole-Parmer, USA). The permeate side consists of a micro gear pump (model: WT3000-1JB, Longer precision pump Co., Ltd. Baoding, China), a storage tank, and a weighing machine (model: 1177 BKWHDR, Resolution: 1 g, Salter Leaf Electronic Digital Scale, FKA Brands Ltd., Tonbridge, Kent, United Kingdom). The permeate side was kept at a constant temperature using a thermostatic bath in circulator mode (LAUDA Alpha RA 8, LAUDA-Brinkmann, LP, Delran, NJ, USA) 

All flow rates were monitored and recorded during the test. The pressure was measured using a pressure meter (HND-P Series, KOBOLD Instruments Inc, Pittsburgh, PA, USA). Feed inlet and outlet temperatures and permeate inlet and outlet temperatures were measured using k-type thermocouples (model: TSK, JUST CO., LTD., Komaki, Japan) and was recorded using an isolated standalone multi-channel datalogger (midi LOGGER GL840, Graphtec Corporation, Yokohama, Japan). Waterproof conductivity meter (CON 150, Eutech Instruments Pte Ltd, Paisley, Scotland) was used to record the total dissolved salts (TDS) and the water conductivity. The feed and permeate flow rates were monitored and directly recorded.

## 3. Results and Discussion

### 3.1. Optimal Concentration of the Carbon Nanotubes Content

As mentioned in Section 2.2, different CNTs concentrations were used for the preparation of PES/CNTs membranes with percentage of 0.1, 0.3, 0.5, 0.7, and 1 wt.% respectively. Homogenous dispersion of the CNTs in the membrane polymer matrix and hydrophobicity of the prepared membranes were considered as the two crucial criteria to predict the enhancement in performance of the prepared membrane to select the membrane CNTs content accordingly. Figure 5 shows that by increasing the CNTs percentage to 0.5 wt.%, the static water contact angle reached a maximum value of 128°, and with further increase in CNTs content contact angle decreased slightly to 125°. This phenomenon can be explained as follows. Higher addition of CNTs content resulted in aggregations which consequently resulted in lowering the contact angle, as well as the poor distribution of the CNTs in the polymer matrix. Therefore, 0.5 wt.% of CNTs were chosen as the additive optimal percentage for all membranes. It is important to mention that although adding CNTs pronounce the thermal conductivity and consequently the thermal polarization since in this study a very small percentage was used (about 0.5%), authors think that the effect of the thermal polarization related to the presence of the CNTs won’t be noticeable.

### 3.2. Membrane Characterization

#### 3.2.1. SEM and TEM Observation

Based on the SEM surface images of the different prepared membranes shown in Figure 6, almost uniform fiber diameters and smooth surfaces were fabricated. The low content of CNTs in the membrane dope composition and sonication of CNTs in the organic solvent led to appropriate distribution of the nanoparticles and prevented CNTs aggregation. These results are in line with the findings of Luong [53]. The diameter of CNTs (110–170 nm) is much thinner than those of membranes nanofibers. Thus, the CNTs seem to be coated by the polymer. ImageJ software was used to estimate the average fiber diameter. The fiber diameters in five membranes were almost the same around 347nm with a maximum 5% deviation that is believed as an experimental error for different dope solutions. Thus, it can be concluded that the fiber diameters were mainly affected by the electrospinning process parameters rather than the membrane composition.

As shown in Figure 7, TEM images for the triple-layer PcH-PES-PcH/CNTs nanocomposite membrane were shown at two different magnifications. Figure 7a shows the homogenous distribution of the CNTs in the membrane matrix. Using higher magnification of the TEM, CNTs were observed in the membrane nanofiber membranes.

#### 3.2.2. Water Contact Angle Measurement

As shown in Figure 8, while pure PES membrane showed the lowest contact angle of 92°±1.7° (Figure 8a), pure PcH membrane contact angle was 119° ± 1.8° (Figure 8c). By adding CNTs to PES, the contact angle increased to 128° ± 1.5° (Figure 8b), which is higher than pure PCH. As it was strongly expected, both PcH/CNTs and triple-layer PcH-PES-PcH/CNTs nanocomposite membranes showed a similar surface contact angle of 144° ± 2° (Figure 8d), because the PES/CNTs layer is sandwiched between two layers of the PcH/CNTs. Higher hydrophobicity (higher contact angle values) is directly related to the presence of CNTs, which was proven to exhibit high hydrophobicity [54]. It is worth mentioning that besides the hydrophobic nature of the CNTs particles, the presence of the CNTs increased the surface roughness which resulted in an increase in hydrophobicity. This is in full agreement with the relationship between roughness and wettability which had been already defined in 1936 by Wenzel [55]. In other words, if the surface is chemically hydrophobic, increasing the surface roughness makes the surface more hydrophobic. For MD applications, it is crucial to create high surface hydrophobicity to avoid membrane wetting, which is the main drawback that sharply decreases the membrane performance. Meanwhile, the hydrophilicity of the membrane bulk material is also a matter of importance to avoid the capillary condensation phenomenon (i.e., avoiding the possibility of water vapor condensation inside the membrane pores structure) that can decrease the MD membrane performance as well [56]. Contact angle results of the triple-layer nanocomposite membrane show noticeably high surface hydrophobicity (water contact angle of 144° ± 2°) with appropriate membrane bulk hydrophobicity, which is expected to be similar to of PES/CNTs. Thus, considering the hydrophobicity of the prepared membrane, it is expected that the triple-layer nanocomposite membrane would be potentially suitable for MD applications.

#### 3.2.3. Membrane Thickness, Porosity, LEP, and Average Fiber and Pore Diameters Measurements

Prepared membranes were characterized in terms of their thickness, porosity, LEP, and average fiber and pore diameters. The results were tabulated in Table 2. The thicknesses of the membranes were hardly changed with different polymeric solution compositions. However, we can notice that adding CNTs to the pure polymers resulted in a slight reduction in the membrane thickness (7.1% reductions in the case of PES/CNTs membrane and 14.8% in case of PcH/CNTs membrane) compared to those of pure PES and pure PcH membranes, respectively. Although a more detailed study is needed, we hypothesize that this decrease in membrane thickness might be related to the excellent conductivity of CNTs, which resulted in higher solution conductivity. This may have increased the attraction forces between the fibers during the electrospinning process, thus resulted in creating thinner layers. 

Prepared membranes showed different bulk porosities. High bulk porosity of the prepared membranes is essential for the prepared MD performance [57]. From Table 2, while the pure PES membrane with 92 ± 1.6% porosity has the highest porosity among all fabricated membranes, PcH membranes with or without CNTs showed the lowest porosity of around 86–87%. Although PES/CNTs showed slightly lower porosity than that of PES, the presence of PES in the triple-layer nanocomposite PcH-PES-PcH/CNTs membrane could positively affect the membrane porosity to be around (91 ± 1.8%). As the higher porosity is suitable to obtain the higher MD performance, the sandwich of the PES layer is essential to increase the MD process performance as well as to decrease the material cost of the prepared membrane. The average pore and fiber diameters are reported in Table 2. 

The LEPs of different prepared membranes are also summarized in Table 2. For pure PES and PES/CNTs, membranes LEP cannot be recorded because both membranes show breakthroughs of the water at lower detectable limits of the used gauge pressure. It might be concluded from here that electrospun modified PES by CNTs membranes are not appropriate for utilization in the MD processes by themselves, because low LEP resulted in membrane wetting, which drastically decreases the membrane performance. The LEP of PcH-PES-PcH/CNTs nanocomposite membrane is similar to that of the PcH/CNTs membrane. Although the reported LEP in this study is somehow lower than that reported in previous studies, for the application we have considered here, it looks that the 1 bar LEP is adequate. Also, the porosity of PcH-PES-PcH/CNTs composite membrane is adequate (91%) due to the existence of the middle PES layer with the highest porosity (92%).

The average fiber and pore diameters and the results were tabulated in Table 2. The results show that, pure PES membrane has the larger average pore diameter and also the added CNTs resulted in reduction in the average pore diameter of both PES/CNTs and PcH/CNTs membranes. Meanwhile, both PcH/CNTs and PcH-PES-PcH/CNTs composite membranes show the same average pore diameter around (0.55 microns). For the fiber diameter, the fibers are mostly homogeneous for each polymeric membrane, with average diameters of 0.342 ± 0.06 µm, 0.366 ± 0.04 µm, 0.334 ± 0.05 µm, 0.338 ± 0.04 µm and 0.343 ± 0.05 µm for PES, PES/CNTs, PcH, PcH/CNTs, and PcH-PES-PcH/CNTs, respectively. These results show that the fiber diameters were mainly affected by the electrospinning process parameters rather than the membrane composition.

Thus, considering using cost-effective materials such as PES modified with CNTs not only resulted in a considerable reduction in the membrane material cost, but also in an increase of the bulk porosity of the triple-layer nanocomposite membrane, which is necessary for the preparation of an efficient MD membrane.

### 3.3. Membrane Distillation Performance Evaluation 

The performance of a direct contact MD system was studied at different feed temperatures, feed flow rates and feed salt concentrations for three different membranes; PcH, PcH/CNTs, and PcH-PES-PCH/CNTs.

#### 3.3.1. Effect of Feed Water Temperature

The feed water temperature was varied from 35 ± 1 °C to 65 ± 1 °C by a step of 10 °C, while the feed inlet conditions were kept constant including the flow rate at 27 L/h and salt concentration of 10,000 ppm. As shown in Figure 9, increasing the feed temperature increased the membrane flux. This is due to the higher vapor pressure difference between feed and permeates sides according to Antoine equation, which expresses the relationship between the liquid temperature and the corresponding equilibrium vapor pressure. Thus, increasing the driving force for water evaporation enhances membrane permeation flux. Triple-layer nanocomposite PcH-PES-PcH/CNTs membrane showed the highest flux of 22.2 kg/m^2^h at a temperature of 65 ± 1 °C, whereas the lowest flux at the same temperature was 11.9 kg/m^2^h for pure PcH membrane. Adding CNTs into PcH resulted in enhancing the permeate flux by 43%. Compared to the PcH membrane, triple-layer nanocomposite PcH-PES-PcH/CNTs membrane showed 83% higher permeation flux. Considering the similar contact angle and LEP for PcH and triple-layer nanocomposite PcH-PES-PcH/CNTs membranes, the higher porosity of the triple-layer membrane looks to be the effective parameter of the membrane, which affects the MD performance. As can be seen in Figure 9, the salt rejection for the triple layer membrane was almost 100% which means that the permeated water was almost pure.

#### 3.3.2. Effect of Feed Water Flow Rate

The effect of the feedwater flow rate on the membrane flux and the salt rejection was studied, and the results were shown in Figure 10. The feed flow rate was varied from 9 to 27 L/h, Reynolds number increased from around 1650 to 4950, while the feed temperature and salt concentration were kept constant at 65 ± 1 °C and 10,000 ppm, respectively. As the feed flow rate increased, the permeate flux increased for all the membranes, PcH, PcH/CNTs, and triple-layer nanocomposite PcH-PES-PcH/CNTs. At the lowest feed flow rate of 9 L/h, the membranes permeation flux was almost identical, since, at low flow rates, the boundary layer effects were predominant. However, at higher flow rates, thus, the boundary layer and temperature polarization effects were reduced due to the turbulence effect [58], and consequently, membrane permeation flux increased for all membranes. A significant enhancement for the triple-layer nanocomposite PcH-PES-PcH/CNTs (164% improvement) is completely clear from Figure 10. It is worth explaining that within the range of the flow rate studied (9 to 27 L/h), the salt rejection decreased from 99.9% to 99.3 %, which indicates that the change in salt rejection was marginal.

#### 3.3.3. Effect of Salt Concentration

The effect of salt concentration against permeate flux is shown in Figure 11. Salt concentration varied from 10,000 to 34,000 ppm while the feed temperature and flow rate were kept constant at 65 ± 1 °C and 27 L/h, respectively. At high feed salt concentration, concentration polarization plays a significant role [59], where a thicker boundary layer reduces the mass transfer driving force. For PcH/CNTs membrane, the permeation flux decreased from 15.3 kg/m^2^h at 10,000 ppm to 5.4 kg/m^2^h (about 64.7% reduction) at 34,000 ppm, while the flux of PcH-PES-PcH/CNTs triple-layer nanocomposite membrane was reduced from 22.2 kg/m^2^h to 8.4 kg/m^2^h (61.8% reduction). As it is clear from Figure 11, at 34,000 ppm, permeation fluxes of the different evaluated membranes are almost the same. It means that at high salt concentrations, it seems that the effect of the membrane material is negligible. We hypothesized that this reduction in flux at a high salt concentration of feed could be attributed to the salt crystallization on the membrane surface, as shown in Figure 12. Indeed, the membrane flux can be recovered by washing the membrane to remove scales from it. 

Although not shown here, the triple-layer PcH-PES-PcH/CNTs nanocomposite membrane showed slightly lower salt rejection than that of the PcH/CNTs membrane (98.7 and 99.9% salt, respectively) at 34,000 ppm feed salt concentration. This tiny difference in salt rejection may be attributed to the higher porosity of the PcH-PES-PcH/CNTs composite membrane that showed much higher permeation flux at the same conditions than the PcH/CNTs membrane as previously showed. 

To have a better understanding of the MD performance of the prepared membranes, the permeation flux is compared with those reported in previously published literature with similar conditions. The comparisons are summarized in Table 3. The prepared triple-layer PcH-PES-PcH/CNTs composite membrane showed comparable characteristics that make it a promising candidate for the MD desalination process. It is clear from Table 3 that permeation fluxes of PcH or composites of PcH and PVDF membranes were less than 7 kg/m^2^h. The triple layer of nanocomposite PcH-PES-PcH/CNTs membrane used in this study showed that the minimum permeation flux was around 8.4 kg/m^2^h, and the maximum one was 22.2 kg/m^2^h. We strongly believe that this increment in MD membrane performance is related to the presence of the middle layer PES with CNTs that had high porosity. Thus, it can be concluded that the newly developed triple-layer PcH-PES-PcH/CNTs composite membrane is appropriate for MD applications.

## 4. Conclusions

The possibility of utilizing modified PES as a cost-effective membrane bulk material for the MD process was evaluated using the electrospinning method. Triple-layer nanocomposite PcH-PES-PcH/CNTs and two control PcH, PcH/CNTs membranes were fabricated using the electrospinning method and their performance in a direct contact MD system was investigated. PES/CNTs showed very low LEP that made it very difficult to apply for MD application. Thus, a PES/CNTs layer was sandwiched between two layers of the PcH/CNTs, which gave very high surface hydrophobicity (water contact angle of 144° ± 2°) and appropriate LEP suitable for the MD applications. PES/CNTs layer hydrophobicity was close to that of the PcH and similar to those membranes vastly used for MD in many published papers. Appropriate hydrophobicity of the bulk structure ensures avoiding capillary condensation inside the membrane, which is important for MD applications. Combining both effects of the high surface hydrophobicity and high bulk porosity resulted in a membrane with adequate MD performance. Permeation flux of the triple-layer nanocomposite membrane reached 22.2 kg/m^2^h with a salt rejection of 99.3% at 27 L/h feed flow rate, 65 ± 1 °C feed temperature, and 10000 ppm salt concentration. This novel three-layer PcH-PES-PcH/CNTs membrane showed superior performance for the MD desalination process with a reasonable cost relative to the other commercial MD membranes.

## Figures and Tables

**Figure 1 membranes-10-00015-f001:**
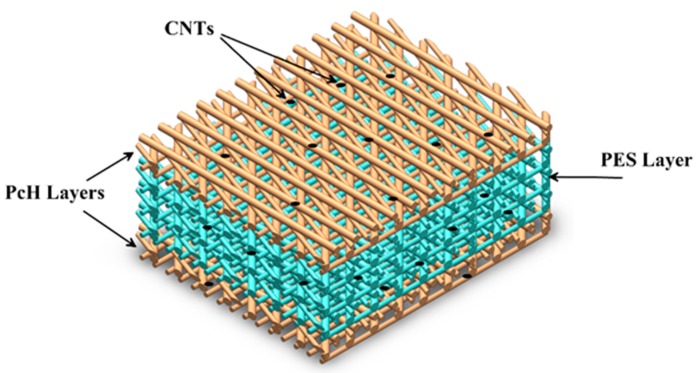
Schematic of the prepared triple-layer nanocomposite membrane.

**Figure 2 membranes-10-00015-f002:**
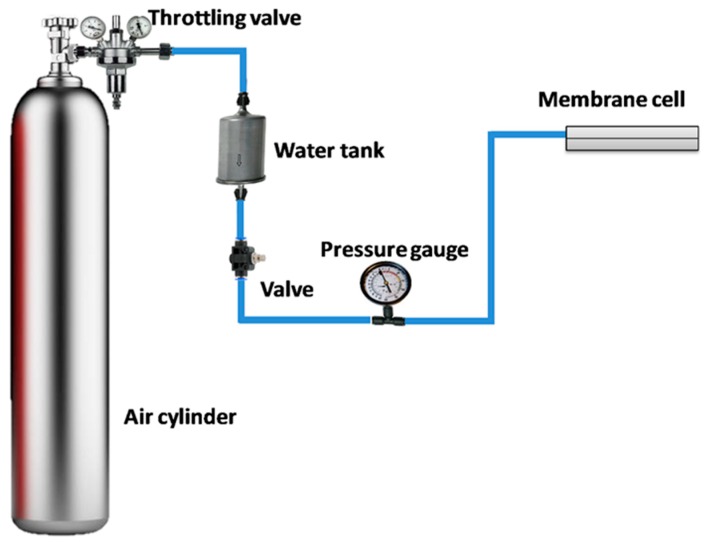
The liquid entry pressure investigation set up.

**Figure 3 membranes-10-00015-f003:**
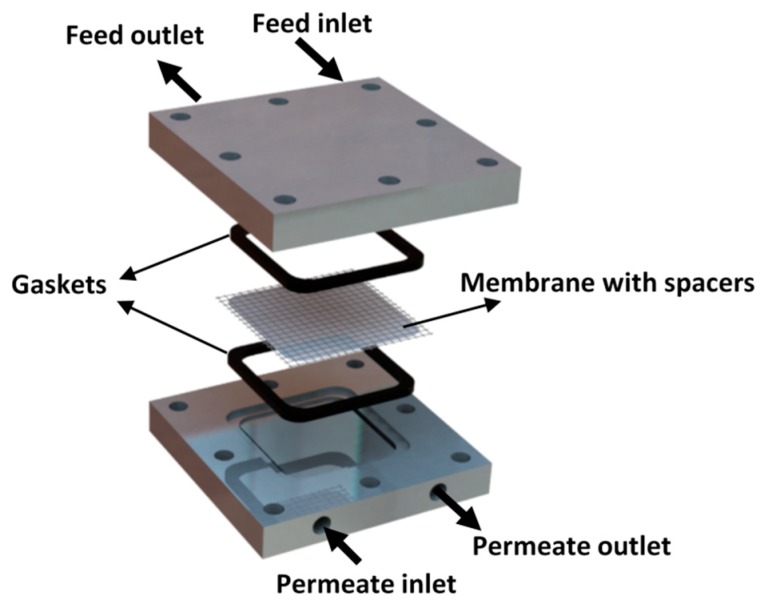
Schematic diagram of the fabricated membrane distillation (MD) cell.

**Figure 4 membranes-10-00015-f004:**
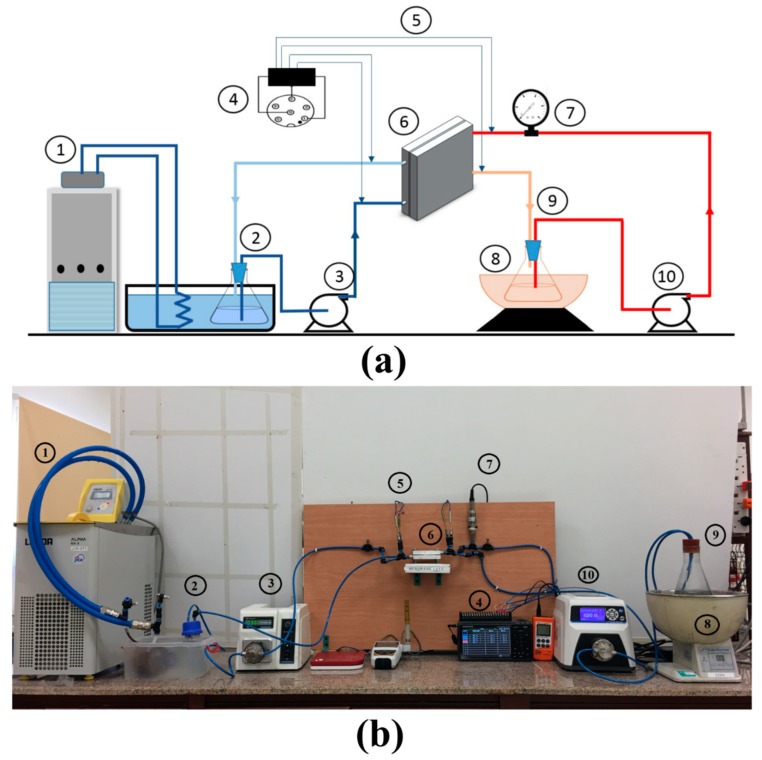
(**a**) Schematic diagram and (**b**) Photo of the used experimental test setup: 1. Thermostatic bath equipped with circulator, 2. Permeate flask, 3. Permeate pump, 4. Data logger, 5. Thermocouples, 6. MD Cell, 7. Pressure meter, 8. Heating bath, 9. Feed flask, and 10. Feed pump.

**Figure 5 membranes-10-00015-f005:**
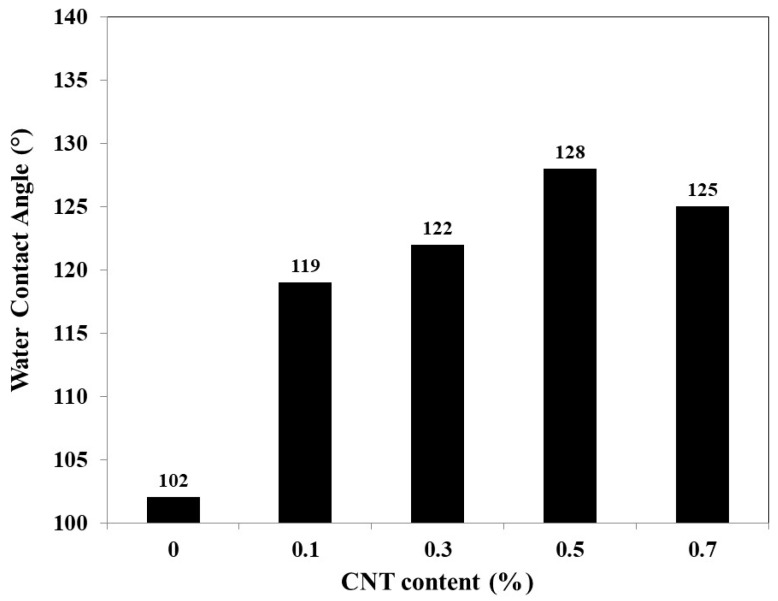
Contact angle measurements.

**Figure 6 membranes-10-00015-f006:**
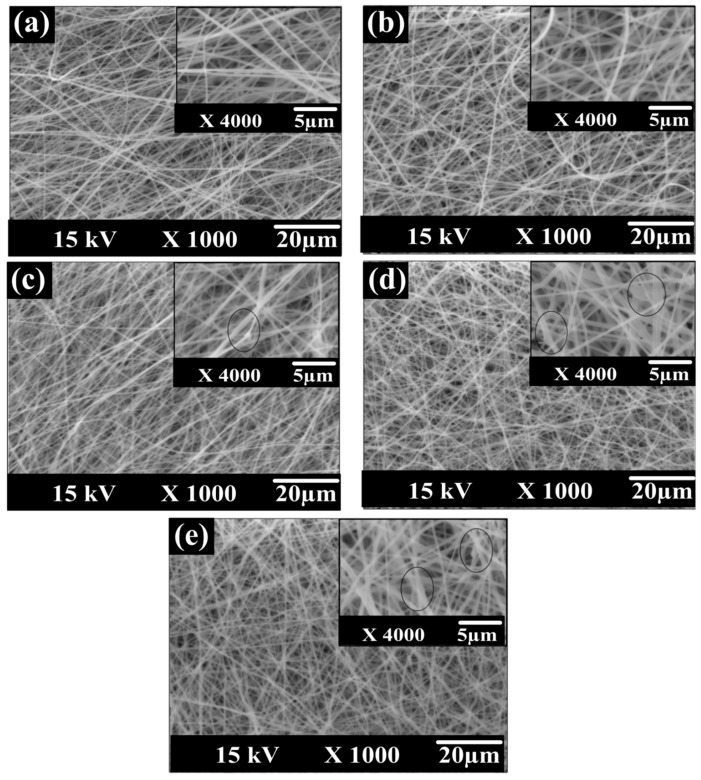
SEM images for the prepared membranes surface: (**a**) pure polyethersulfone (PES), (**b**) PES/carbon nanotubes (CNTs), (**c**) Pure poly (vinylidene fluoride-co-hexafluoro propylene (PcH), (**d**) PcH/CNTs, and (**e**) triple-layer PcH-PES-PcH/CNTs nanocomposite membranes.

**Figure 7 membranes-10-00015-f007:**
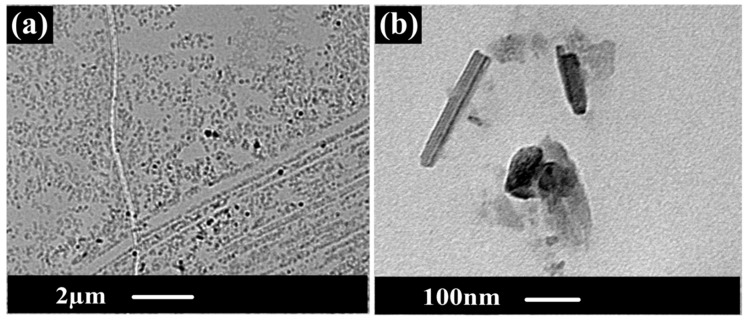
TEM cross-section images for the PcH-PES-PcH/CNTs membrane at different scales. (**a**) shows the homogenous distribution of CNTs; (**b**) the average size of CNTs.

**Figure 8 membranes-10-00015-f008:**
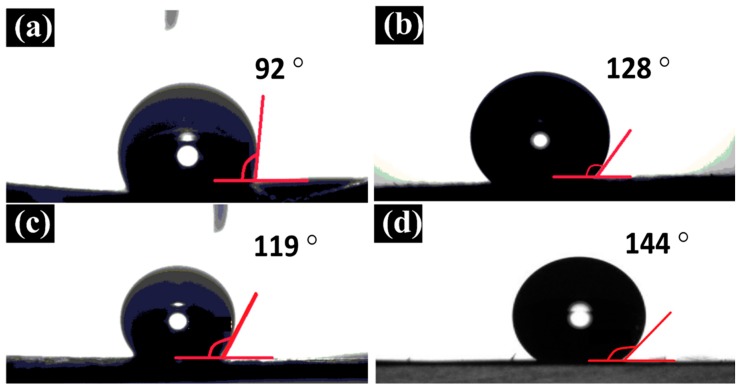
Contact angle results for (**a**) pure PES, (**b**) pure PcH, (**c**) PES/CNTs, and (**d**) PcH-PES-PcH/CNTs, which is almost the same as PcH/CNTs.

**Figure 9 membranes-10-00015-f009:**
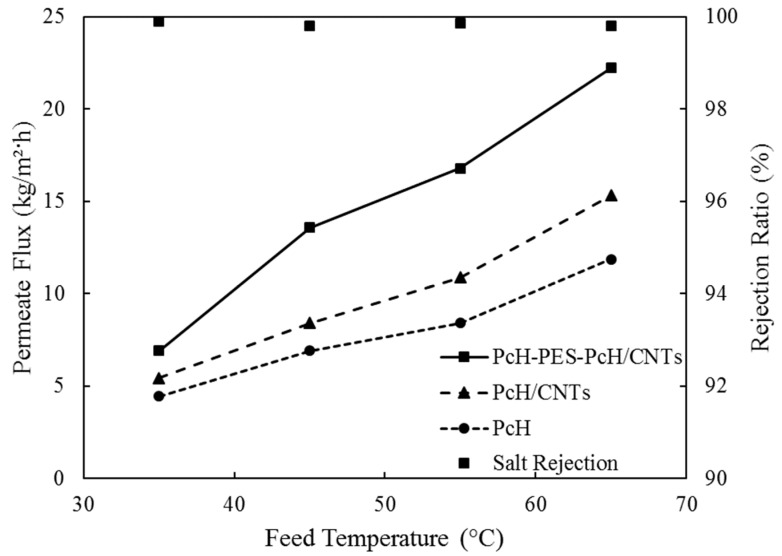
Permeate flux vs. feed water temperature for different membranes.

**Figure 10 membranes-10-00015-f010:**
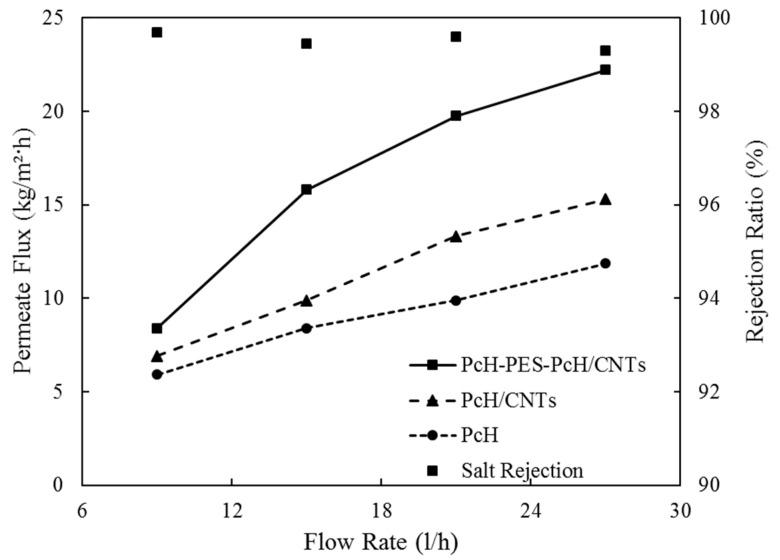
Permeate flux vs. feedwater flow rate for different membranes.

**Figure 11 membranes-10-00015-f011:**
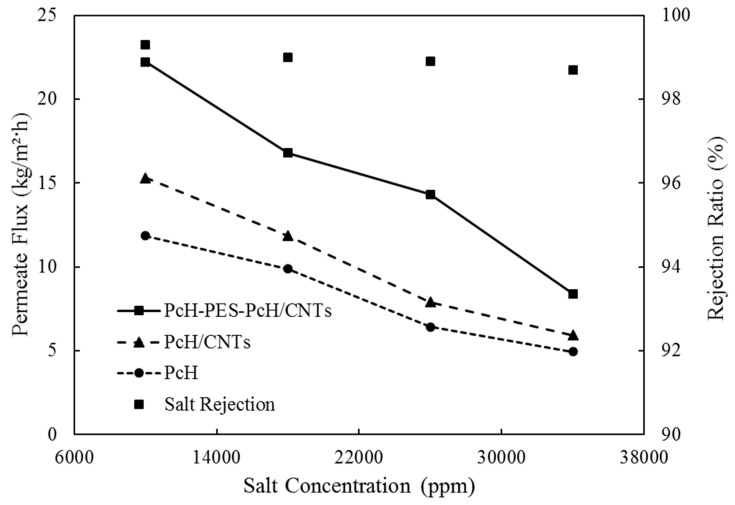
Permeate flux vs. feed salt concentration for different membranes.

**Figure 12 membranes-10-00015-f012:**
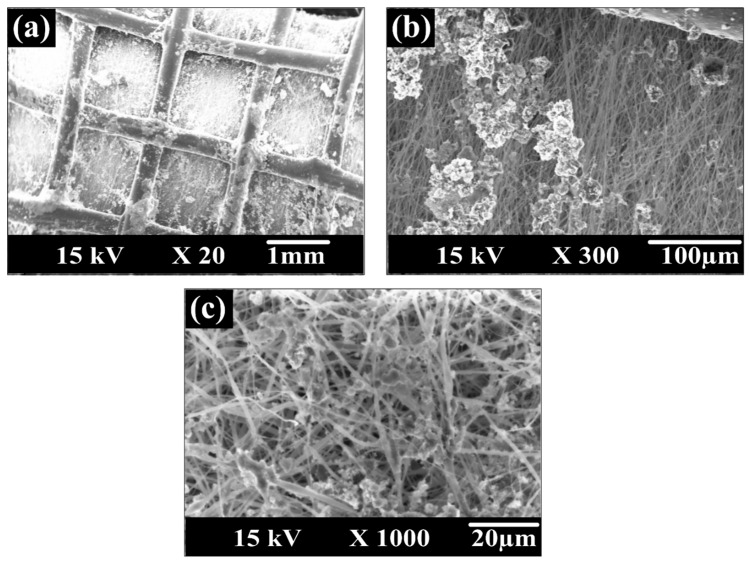
SEM images of the formed scales on (**a**) the spacer, (**b**) and (**c**) PcH-PES-PcH/CNTs nanocomposite membrane fibers, and with higher magnifications.

**Table 1 membranes-10-00015-t001:** Codes and compositions of the prepared membranes.

Membrane Code	PES wt.%	PcH wt.%	CNTs wt.%
PES	23	-	-
PcH	-	15	-
PES/CNTs	23	-	0.5
PcH/CNTs	-	15	0.5
PcH-PES-PcH/CNTs	23	15	0.5

**Table 2 membranes-10-00015-t002:** Characteristics of different produced membranes include liquid enter pressure (LEP), membrane thickness, membrane porosity, contact angle, and the membrane fiber diameter.

Membrane Code	Membrane Thickness (µm)	Porosity ε (%)	LEP * (bar)	Average Fiber Diameter (µm)	Average Pore Diameter (µm)
PES	112 ± 2.1	92 ± 1.6	NA	0.342 ± 0.06	0.79 ± 0.05
PcH	108 ± 2.3	87 ± 1.5	1.7	0.366 ± 0.04	0.67 ± 0.05
PES/CNTs	104 ± 1.9	89 ± 2.0	NA	0.334 ± 0.05	0.63 ± 0.03
PcH/CNTs	92 ± 2.3	86 ± 2.1	1.8	0.338 ± 0.04	0.56 ± 0.04
PcH -PES-PcH/CNTs	107 ± 2.8	91 ± 1.8	1.8	0.343 ± 0.05	0.55 ± 0.03

* LEP: liquid enter pressure (absolute).

**Table 3 membranes-10-00015-t003:** Characteristics and performance of the fabricated PcH-PES-PcH/CNTs composite membrane in this study compared to those of previously reported membranes.

Polymer	Membrane Characteristics	Feed Concen.	T_feed_ (°C)	Temperature (°C)	Flux (kg/m^2^h)	Ref.
Silica-PVDF composite, electrospinning	Contact angle 156.3°	3.5 wt.% NaCl	60	20	18.1	[17]
PcH-CNTs	Contact angle 91.65°	3 wt.% mixed salts	74	56	0.6–1	[60]
PVDF-based membranes, electrospinning	Contact angle 128°–154°	1–3.5 wt.% NaCl	50–80	17–24	4.28	[61]
TiO_2_ and fluoro-silane compound coating on PTFE	Average pore diameter: 0.25 μm	Up to 10 wt.% NaCl	60	25	4–6	[62]
PVDF-PcH	contact angle: 96.4°	5 wt.% NaCl solution	80	17	9	[63]
PcH-CNF	contact angle: 115°	3 wt.% NaCl	70	20	7-8	[64]
PcH-PES-PcH/CNTs composite membrane	Porosity: 91 ± 1.8%; contact angle:144° ± 2°	1–3.5 wt.% NaCl solution	35–65	20-25	8.4–22.2	This study

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
