# Peer review of "Triple-Layer Nanocomposite Membrane Prepared by Electrospinning Based on Modified PES with Carbon Nanotubes for Membrane Distillation Applications"

_membranes, 2020, doi:10.3390/membranes10010015_

Round 1

Reviewer 1 Report

The manuscript is written well and scientifically sound.

There are just a few points that need the author's attention

L107: "PES or modified PES has rarely been applied for MD applications" There are studies that assessed PES for example:https://onlinelibrary.wiley.com/doi/abs/10.1002/app.45516, therefore, the authors are requested to re-write the statement Table 2: LEP is measured in absolute values. That means atm is considered as 1 bar right? If its true the LEP of the fabricated membranes is low. As PVDF and PTFE membranes have around 2 bar LEP. So these membrane do not perform as well as classical PVDF and PTFE membranes also longer tests were not done therefore a reader cannot see how these membrane perform as they age.
L337: Can the authors report the values of flow rate in Reynolds number inside the membrane cell in the entire manuscript. As flow rates in a particular cell can be high or low depending on the cell design and these results cannot be interpreted or used by other researchers. Fig 9: Can the authors put the salt rejection values for the three membranes on the secondary axis to see how these membranes are performing w.r.t wetting L364: Remove "utterly"

Author Response

Response to the comments of the reviewers

First of all, we, the authors, would like to express our deepest gratitude for the Journal editors and all the distinguished reviewers for their kind efforts and great reviews comments that benefited us in improving our manuscript quality. The response to the comments of the reviewers is given below; the original comment is given in bold black, while the reply is given right below in normal print in blue color. The changes in the manuscript were highlighted by red color.

Reviewer #1:

There are just a few points that need the author's attention

[Comment 1] L107: "PES or modified PES has rarely been applied for MD applications" There are studies that assessed PES for example: https://onlinelibrary.wiley.com/doi/abs/10.1002/app.45516, therefore, the authors are requested to re-write the statement

[Answer]  The valued reviewer pointed out that the PES was used before in MD applications. The authors would like to thank the reviewer for informing us about this important point. The statement was corrected in the manuscript and the reference was added.

Based on the valued reviewer comment the following explanations were added on page 3, line L105 to L107. The revised parts were highlighted in red color.

Up till now, PES has rarely been applied for MD applications, as it is not inherently hydrophobic, although it has been widely used in porous membranes (MF and UF) preparation. Few studies investigated the utilization of phase-inversion-coated PES membranes in MD [49,50].

Eykens, L.; De Sitter, K.; Stoops, L.; Dotremont, C.; Pinoy, L.; Van der Bruggen, B. Development of polyethersulfone phase-inversion membranes for membrane distillation using oleophobic coatings. Journal of Applied Polymer Science 2017, 134, 45516, doi:10.1002/app.45516. Rastegarpanah, A.; Mortaheb, H.R. Surface treatment of polyethersulfone membranes for applying in desalination by direct contact membrane distillation. Desalination 2016, 377, 99-107, doi:https://doi.org/10.1016/j.desal.2015.09.008.

 [Comment 2] Table 2: LEP is measured in absolute values. That means atm is considered as 1 bar, right? If it’s true the LEP of the fabricated membranes is low. As PVDF and PTFE membranes have around 2 bar LEP. So, this membrane does not perform as well as classical PVDF and PTFE membranes also longer tests were not done therefore a reader cannot see how this membrane perform as they age.

[Answer]  The authors acknowledge the valued reviewer comments. The reviewer has commented about the lower LEP of the membranes in this study compare to those ones reported in the literature. Basically, authors do agree well with the raised comment with the valued reviewer that the LEP of membranes prepared in this study is low due to the large average pore diameter of the prepared membranes via electrospinning technique. Although LEP values are relatively low, in our opinion they are adequate for MD applications, as the maximum vapor pressure difference across membrane surfaces is around 0.16 bar.

Based on the valued reviewer comment the following explanations and sentences were added on page 11, line  327 to 329. The revised parts were highlighted in red color.

Although the reported LEP in this study is somehow lower than that reported in previous studies, for the application we have considered here, it looks that the 1 bar LEP is adequate.

[Comment 3] L337: Can the authors report the values of the flow rate in the Reynolds number inside the membrane cell in the entire manuscript. As flow rates in a particular cell can be high or low depending on the cell design and these results cannot be interpreted or used by other researchers.

[Answer]  The valued reviewer has addressed the importance of adding Reynold's number. The authors acknowledge the importance of Re for the purpose of the accurate comparison of the obtained results from other research group's results.  

Based on the valued reviewer comment the Reynolds number values were added to the manuscript on Page 13, line 371 to 372. The revised parts were highlighted in red color.

Reynolds number increased from around 1650 to 4950.

[Comment 4] Fig 9: Can the authors put the salt rejection values for the three membranes on the secondary axis to see how these membranes are performing w.r.t wetting.

[Answer]  The valued reviewer has kindly requested to include the slat rejection in Fig. 9.

Based on the valued reviewer comment the salt rejection values were added to Fig. 9 as requested by the valued reviewer on page 13 and the revised part was added on page 12, lines 364 to 366. The revised parts were highlighted in red color.

As can be seen in Figure 9, the salt rejection for the triple-layer membrane was almost 100% which means that the permeated water was almost pure

[Comment 5] L364: Remove "utterly"

[Answer] Authors do agree with the comment raised by the valued reviewer. The word "utterly" was removed from the manuscript on page 13 line 370.

Reviewer 2 Report

The manuscript titled "Triple-layer nanocomposite membrane prepared by electrospinning based on modified PES with carbon nanotubes for membrane distillation application" described the prepared nanocomposite hydrophobic membranes for desalination. The author chose PES as bulk membrane material and modified it with PcH/CNTs to obtain a high porosity triple-layer membrane by electrospinning method. The optimal concentration of CNTs was confirmed and the performance differences of several prepared membranes were also compared. However, there are still some work should be strengthened.

(1) In Section 3.2.1, I think it would be better if the diameter data of nanofiber can be provided. And for Figure 7(a), the whole cross-section picture of membrane should be provided.

(2) In Section 3.3.1, because of the great thermal conductivity of CNTs, the temperature adhere to membrane surface would be lower than the bulk feed solution. Is it necessary for us to take temperature polarization into account?

(3) As for MD process, the mass transfer resistance of triple-layer would be high than the one layer. Though the porosity listed in Table 2 was not much different, it is better to provide the data about pore size of all prepared membranes.

(4) In Section 3.2.2, “However, at higher flow rate ….. due to the turbulence effect”, for the flowing state of feed solution, we need to calculate the Reynolds number according to your own experimental condition not from the reference.

(5) MD process should be a long-term operation, and also can be considered as concentration. Therefore, the concentration of saline solution would must be higher and higher. But in Section 3.2.3, “Although not shown here, … , at 34000 ppm feed salt concentration.” In fact, the salt rejection of triple-layer membrane would be lower than one layer. Therefore, author should add the long-term stability experiment to support the excellent performance of triple layer membranes.

Author Response

Response to the comments of the reviewers

First of all, we, the authors, would like to express our deepest gratitude for the Journal editors and all the distinguished reviewers for their kind efforts and great reviews comments that benefited us in improving our manuscript quality. The response to the comments of the reviewers is given below; the original comment is given in bold black, while the reply is given right below in normal print in blue color. The changes in the manuscript were highlighted by red color.

Reviewer #2:

The manuscript titled "Triple-layer nanocomposite membrane prepared by electrospinning based on modified PES with carbon nanotubes for membrane distillation application" described the prepared nanocomposite hydrophobic membranes for desalination. The author chose PES as bulk membrane material and modified it with PcH/CNTs to obtain a high porosity triple-layer membrane by electrospinning method. The optimal concentration of CNTs was confirmed and the performance differences of several prepared membranes were also compared. However, there are still some work should be strengthened.

[Comment 1] In Section 3.2.1, I think it would be better if the diameter data of nanofiber can be provided. And for Figure 7(a), the whole cross-section picture of the membrane should be provided.

[Answer] The valued reviewer has requested to provide the nanofiber diameter should be provided. We do agree with the valued reviewer The average nanofibers' diameters data are necessary.

Based on the valued reviewer comment the nanofibers' average diameters were added to the manuscript on page 12 table 2 column 5. The revised parts were added on page 5 lines from 194 to 196 and highlighted in red color. The revised parts were added on pages 11 and 12 lines from 331 to 339 and highlighted in red color.

As another point, the valued reviewer has pointed out to provide the whole cross-section SEM image of the prepared membranes in Fig.7.  We kindly would like to ask the distinguished reviewer's attention to the following points:

in Fig. 12(b) we have provided almost a whole cross-section of the membrane (please kindly consider the thickness of the membranes that we have reported) and discussions based on the cross-section structure were already done for Fig. 12b. As a matter of fact, Fig. 7 is a TEM image. The authors tried to include some analysis for the explanation of the phenomena. Including SEM image of the cross section with very low magnification (to show the whole cross-section), might affect the logical flow of the concepts in the manuscript. I mean it would be difficult to discuss about TEM and requested SEM image as one figure. Since the structure on both sides is the same (symmetric structure), the whole cross-section can be obtained in very low magnification that is not so much suitable for accurate observation of the structures. Thus, observing half of the membrane thickness with higher magnification might be enough. We have provided several SEM images in different parts to support our expected phenomena based on the membrane structure that we need. Since almost a whole cross-section was provided in 12 (b), and considering point 3, it looks that the membrane structure was evaluated.

Thus, the authors think that considering the above mentioned points, the SEM image of the Fig. 12 (b) is good enough especially if we consider the symmetric structure of the membrane.

"ImageJ software was used to investigate the average pore size and the average fiber diameter of the prepared membranes [25,51,52]. Four hundred pores were considered using images to determine the average pore diameter of the membranes using image processing."

"The average fiber and pore diameters and the results were tabulated in Table 2. The results show that, pure PES membrane has the larger average pore diameter and also the added CNTs resulted in a reduction in the average pore diameter of both PES/CNTs and PcH/CNTs membranes. Meanwhile, both PcH/CNTs and PcH-PES-PcH/CNTs composite membranes show the same average pore diameter around (0.55 microns). For the fiber diameter, the fibers are mostly homogeneous for each polymeric membrane, with average diameters of 0.342±0.06 µm, 0.366±0.04 µm, 0.334±0.05 µm, 0.338±0.04 µm and 0.343±0.05 µm for PES, PES/CNTs, PcH, PcH/CNTs, and PcH-PES-PcH/CNTs, respectively. These results show that the fiber diameters were mainly affected by the electrospinning process parameters rather than the membrane composition."

[Comment 2] In Section 3.3.1, because of the great thermal conductivity of CNTs, the temperature adhere to membrane surface would be lower than the bulk feed solution. Is it necessary for us to take temperature polarization into account?

[Answer] The authors acknowledge valued reviewer comments. We totally agree that theoretically, temperature polarization (TP) phenomenon would be more likely to occur in the case of using CNTs, due to the increase in the total membrane thermal conductivity. We would like to drag the attention of the valued reviewer to this point that the ratio of the CNTs is very small (about 0.5 %), the expected effect is very low and can be neglected in favor of other benefits created due to the existence of the CNTs.

Based on the valued reviewer comment the following explanations were added in page 7 and 8  Line 250 to 253.

It is important to mention that although adding CNTs pronounce the thermal conductivity and consequently the thermal polarization since in this study a very small percentage was used (about 0.5%), authors think that the effect of the thermal polarization related to the presence of the CNT won't be high.

[Comment 3] As for the MD process, the mass transfer resistance of the triple-layer would be high than the one layer. Though the porosity listed in Table 2 was not much different, it is better to provide the data about the pore size of all prepared membranes.

[Answer] Authors completely do agree with the valued reviewer that the pore size of the prepared membranes should be added in the manuscript. Thus, based on the valued reviewer comment, the average pore diameter of the prepared membranes was added to the manuscript in page 12, table2 Column 6. The revised parts were added in page 5 lines from 194 to 196 and highlighted in red colour. The revised parts were added in page 11 and 12 lines from 331 to 339 and highlighted in red colour.

"ImageJ software was used to investigate the average pore size and the average fiber diameter of the prepared membranes [25,51,52]. Four hundred pores were considered using SEM images to determine the average pore diameter of the membranes using image processing."

"The average fiber and pore diameters and the results were tabulated in Table 2. The results show that, pure PES membrane has the larger average pore diameter and also the added CNTs resulted in reduction in the average pore diameter of both PES/CNTs and PcH/CNTs membranes. Meanwhile, both PcH/CNTs and PcH-PES-PcH/CNTs composite membranes show the same average pore diameter around (0.55 microns). For the fiber diameter, the fibers are mostly homogeneous for each polymeric membrane, with average diameters of 0.342±0.06 µm, 0.366±0.04 µm, 0.334±0.05 µm, 0.338±0.04 µm and 0.343±0.05 µm for PES, PES/CNTs, PcH, PcH/CNTs, and PcH-PES-PcH/CNTs, respectively. These results show that the fiber diameters were mainly affected by the electrospinning process parameters rather than the membrane composition."

Carlos Mierzwa, J.; Vecitis, C.D.; Carvalho, J.; Arieta, V.; Verlage, M. Anion dopant effects on the structure and performance of polyethersulfone membranes. Journal of Membrane Science 2012, 421-422, 91-102, doi:https://doi.org/10.1016/j.memsci.2012.06.039. Islam, M.S.; McCutcheon, J.R.; Rahaman, M.S. A high flux polyvinyl acetate-coated electrospun nylon 6/SiO2 composite microfiltration membrane for the separation of oil-in-water emulsion with improved antifouling performance. Journal of Membrane Science 2017, 537, 297-309, doi:https://doi.org/10.1016/j.memsci.2017.05.019.

[Comment 4] In Section 3.2.2, “However, at higher flow rate …... due to the turbulence effect”, for the flowing state of feed solution, we need to calculate the Reynolds number according to your own experimental condition not from the reference.

[Answer] The authors acknowledge the valued reviewer comment. The authors acknowledge the importance of Re number in determining the flow characteristics in the MD cell.  Reynolds number values were added to the manuscript.

Based on the valued reviewer comment the Reynolds number values were added to the manuscript in Page 13, line 371,372. The revised parts were highlighted in red color.

Reynolds number increased from around 1650 to 4950,

[Comment 5] MD process should be a long-term operation, and also can be considered as concentration. Therefore, the concentration of the saline solution would must be higher and higher. But in Section 3.2.3, “Although not shown here, …, at 34000 ppm feed salt concentration.” In fact, the salt rejection of triple-layer membrane would be lower than one layer. Therefore, author should add the long-term stability experiment to support the excellent performance of triple layer membranes.

[Answer] We appreciate the valued reviewer's comment about the long-term operation. We acknowledge the importance of long-term stability experiments in determining the true value of the produced membrane, especially for evaluating the potential for reall application. We kindly drag the valued reviewer's attention to this point that, this research as our early try attempt, mainly focused on the fabrication and characterization of the newly developed membrane and its potential in MD operations. The DCMD experiments were added as a proof of concept, that this membrane is suitable for MD applications.

Meanwhile, we strongly acknowledge and do agree with the valued reviewer comment, the authors will have a much more in-depth investigation of the prepared membrane performance including, but not limited to, long-term performance stability experiments. Thus, we appreciate the distinguished reviewer comment and keep strongly this in mind and hopefully, te long-term, stability with other performance evaluations of the prepared membranes will be reported in our further coming manuscript soon.

Round 2

Reviewer 1 Report

Accept in the present form

Reviewer 2 Report

The manuscript titled "Triple-layer nanocomposite membrane prepared by electrospinning based on modified PES with carbon nanotubes for membrane distillation application" has been modified greatly according to the reviewer’s comments. Therefore, I agreed that this manuscript would be published.